# Comparative Metabolic Profiling of Grape Skin Tissue along Grapevine Berry Developmental Stages Reveals Systematic Influences of Root Restriction on Skin Metabolome

**DOI:** 10.3390/ijms20030534

**Published:** 2019-01-28

**Authors:** Shuyan Duan, Yusen Wu, Ruifeng Fu, Lei Wang, Yujin Chen, Wenping Xu, Caixi Zhang, Chao Ma, Jianxin Shi, Shiping Wang

**Affiliations:** 1Department of Plant Science, School of Agriculture and Biology, Shanghai Jiao Tong University, 800 Dongchuan Road, Minhang District, Shanghai 200240, China; hongloudsy@sjtu.edu.cn (S.D.); senwy886@sjtu.edu.cn (Y.W.); leiwang2016@sjtu.edu.cn (L.W.); sjtucyj@sjtu.edu.cn (Y.C.); wp-xu@sjtu.edu.cn (W.X.); acaizh@sjtu.edu.cn (C.Z.); chaoma2015@sjtu.edu.cn (C.M.); 2Joint International Research Laboratory of Metabolic & Developmental Sciences, School of Life Sciences and Biotechnology, Shanghai Jiao Tong University, 800 Dongchuan Road, Minhang District, Shanghai 200240, China; furuifeng0622@sjtu.edu.cn; 3Institute of Agro-food Science and Technology/Key Laboratory of Agro-products Processing Technology of Shandong, Shandong Academy of Agricultural Sciences, Jinan 250100, China

**Keywords:** *Vitis vinifera*, veraison, skin tissue, metabolomics, root restriction

## Abstract

This research aimed to comparatively evaluate the influences of root restriction (RR) cultivation and traditional cultivation (RC) on grape berry skin metabolomics using a non-targeted metabolomics method. Two-hundred-and-ninety-one metabolites were annotated and the kinetics analyses showed that berry skin metabolome is stage- and cultivation-dependent. Our results showed that RR influences significantly the metabolomes of berry skin tissues, particularly on secondary metabolism, and that this effect is more obvious at pre-veraison stage, which was evidenced by the early and fast metabolic shift from primary to secondary metabolism. Altogether, this study provided an insight into metabolic adaptation of berry skin to RR stress and expanded general understanding of berry development.

## 1. Introduction

As a non-climacteric and important economical fruit, grape (*Vitis vinifera* L.) is cultivated worldwide, which possesses approximately 7,400,000 ha in the world [1]. Besides being consumed as table fruits, grape berries are also processed to wine and raisins, which possess a range of health benefits, such as atherosclerosis prevention [2], anti-oxidation [3,4], and renal damage prevention [5]. The development of grape berries consists of two successive sigmoid growth phases separated by an intermediate lag phase, and each phase shows distinctive characteristics in physiology and biochemistry, including a berry’s size and shape, changes in color, texture, and metabolic dynamics [6]. The first phase (berry development phase) begins with pericarp cell division and cell enlargement as principal organic compounds accumulate; the malic acid, tartaric acid, and tannins are especially critical to wine quality [7,8]. The second phase (berry ripening phase) is characterized by the berries’ coloring and softening with a significant reduction in acids concentration due to the enlargement of berry volume and the tremendous increase in sugar compounds. Beyond sugar accumulation, a grape or wine’s quality is mainly determined by the secondary metabolites [7]. For most red grape varieties, anthocyanin accumulation is considered to be the most obvious production during berry ripening, which is restricted to skin tissue in most red grape cultivars [9]. 

The taste and quality of grape and wine generally reflects the compositions of amount of primary and secondary metabolites. Organic acids, amino acids, and sugars are the major primary metabolites, which accumulate mainly in the pulp tissues during the berry formation phase. But most of the secondary metabolites, such as phenylpropanoids include phenolic acids, flavonoids, viniferins, and stilbenes are typically found in the skin tissues during the berry ripening phase [10,11]. A range of biotic and abiotic stresses including water stress, heat stress, solar irradiance, and pathogen infection can have great effects on grape and/or wine quality and compositions [11,12,13,14]. Root restriction (RR) is considered to be another type of stress for crop plants, which has direct and indirect effects on the morphological and physiological properties. Meanwhile, RR is a practical tool for improving the quality of crop plants and the volume utilization efficiency by restricting a plant’s rooting volume available [15]. Root restriction has been well applied to various crop plants, such as cotton (*Gossypium hirsutum* L.) [16], pepper (*Capsicum annuum cv Bellboy*) [17], apple (*Malus pumila*) [18], sweet potato (*Ipomoea batatas* (L.) Lam.) [19], and especially in grape (*Vitis vinifera* L.) [20,21,22,23]. Root restriction was proved to improve the absorbing ability of grapevine root system with more secondary and fabric roots [23,24]. Besides, it is achievable to enhance the nitrate uptake rate and to shape the overall sensory experience of grape berries with increasing concentration of sugars and anthocyanins under RR cultivation [22,24,25,26]. It has been reported that the anthocyanin levels in “Summer black” berry skin were significantly increased in response to RR, which resulted mainly from the increase of tri-hydroxylated, methoxylated, and monoglycosylated anthocyanins [27]. In addition, RR can significantly increase ascorbic acid (AsA) contents that are often used as an index of fruit health-related quality [28].

The release of a draft whole-genome sequence of grapevine in 2007 provides a high-throughput and more insightful method to study the berry development [29]. Recently, a large range of transcriptomic [30], proteomic [31,32], and metabolomic [33,34] studies were performed to reveal the molecular mechanisms underlying grape berry development under normal and stressed conditions. Although transcriptomic studies have revealed the transcriptomic changes during grape berry development under RR condition [30], the simultaneously generated metabolites as the genome, transcriptome, and proteome final regulatory products could not be simply deduced based on the above data. In addition, previous studies demonstrated that metabolic changes could also affect the gene transcript levels [35]. Moreover, metabolites fingerprinting provides a valuable tool to identify the resistance gene responsible for defensing a soil-borne vascular pathogen *Verticillium longisporum* infection in *Arabidopsis* [36]. These results suggest that metabolic analysis will make great contributions to identify novel metabolic makers and pathways involved in plant–environment interactions. However, although metabolomic studies have been carried out to reveal the effects of stresses on grape berry development processes [11,37], and even RR is reported to be useful for improving color and taste quality in grapevine [22], metabolomic study of RR on grape berry development is not reported yet. 

Previous studies demonstrated that significant metabolic changes in developing grape berries occur around the pre-version stage, as evidenced by the decomposition of primary metabolites and the simultaneous accumulation of secondary metabolites [34]. In this study, a non-targeted metabolomics approach was applied to comparatively investigate the effect of RR on dynamic metabolomics in skin tissues of developing “Red Alexandria” grape berries. 

## 2. Results

### 2.1. Metabolite Profiling of Berry Skin Samples

In order to compare the berry skin metabolic kinetic change patterns along berry development between RR and RC, samples of “Red Alexandria” grape berry at eight WAFB (eight weeks after full bloom, S1), 10 WAFB (S2), 12 WAFB (S3), 14 WAFB (S4), 16 WAFB (S5) from RR and RC were collected, and the metabolic profiling was performed using an untargeted ultra high-performance liquid chromatography-mass spectrometry (UHPLC-MS) system. Two-hundred-and-ninety-one metabolites in total were annotated from more than 1000 characteristic features, including 158 primary metabolites (51 amino acids and derivatives, 48 carbohydrates and organic acids, 29 lipids, 19 nucleotides, and 11 CPGECs-cofactors, prosthetic groups, and electron carriers), 114 secondary metabolites (including 73 flavonoids and 41 other phenolics), and 19 other compounds (Appendix A). Notably, six metabolites (glyceraldehyde, ellagic acid hexoside, malvidin 3-*O*-(6-*O*-coumaryl)-glucoside, tricin *O*-glucoside *O*-guaiacylglyceryl ether, uridine, and isorhamnetin) were not found in S1 samples of RC, while one metabolite (tricin *O*-glucoside *O*-guaiacylglyceryl ether) was not found in both S4 and S5 samples of RR and RC.

### 2.2. Kinetic Patterns of Developing Grape Berry Skin Metabolomes

In order to view the kinetic metabolomes of developing grape berry skin, the unsupervised multivariate data analysis of principal component analysis (PCA) was subsequently performed with the annotated 291 metabolites (Figure 1). During grape berry development, the separation between different development stages of the same cultivation methods or between different cultivation methods of the same development stages was clear, revealing a cultivation method and developmental stage dependent metabolic kinetics of developing grape berry skin. The first two principal components (PCs) explained 54.7% of the total variance of the skin metabolism (40% and 14.7% for PC1 and PC2, respectively). Principal Component 1 separated the variations by developmental stages, while PC2 by cultivation method. According to loading plots (Appendix A), amino acids and flavonoids contributed the most to the negative and the positive values of PC1, respectively. Glucose and glycerate also had higher eigenvalues on PC1, indicating their strong contributions to sample distribution along this component. Interestingly, the separation of two cultivation methods in PCA plots at the same stage of development became farther as the grape berry developed, indicating that RR affects grape berry skin metabolome in a stage-dependent way: the more advanced the grape berry development, the bigger the effect.

### 2.3. Metabolic Changes of Grape Berry Skin along Grape Berry Development

To discover the metabolic variations in developing grape berry skins, the levels of the 291 annotated metabolites in a given stage of RR or RC were compared with the corresponding hard-core stage (S1 in RR and RC), respectively, and submitted to clustering analysis to authenticate the stage-dependent variation (Appendix A). *T*-test and false discovery rate (FDR) analysis were used to discriminate.

For primary metabolites in RC (Appendix A), more than 47% (26 at S2, 37 at S3, 33 at S4, and 37 at S5) detected amino acids and their derivatives increased dramatically from the lowest level at S1 to the highest level at S5 along berry development. Most carbohydrates, CPGECs, and nucleotides in RC showed similar increasing trends as amino acids. However, levels of 75.9% (22 out of 29) lipids decreased, and those of 67% (eight out of 12) organic acids kept continually stable along berry development. In RR, changes of primary metabolites along berry development were quite similar with minor differences, which will be further elucidated below. For secondary metabolites in RC (Appendix A), levels of eight to 34 flavonoids increased while those of 21 to 27 flavonoids decreased dramatically along berry development, respectively. In addition, levels of four benzene and substituted derivatives, four hydrolysable tannins, nine cinnamic acids and derivatives, and two others metabolites declined along berry development. Changes of secondary metabolites in RR along berry development were quite different, which will be further presented below.

### 2.4. Metabolic Differences between RR and RC Samples

To investigate the RR effects on metabolome profile, first, a heatmap of metabolite ratios was constructed between RR and RC in each sampling time at different berry development stages (Appendix A). The results showed that the difference of metabolites between RR and RC were stage- and cultivation-dependent. Further analysis focused mainly on the comparison of the effects of RR and RC on primary (Figure 2) and secondary metabolism (Figure 3). 

For primary metabolism, as compared with those in RC, levels of metabolites in glycolysis, such as glucose, glucose-6-phosphate, and fructose-6-phosphate increased dramatically, while those of glycerate and pyruvate declined in RR (Figure 2A). The intermediate metabolites in tricarboxylic acid cycle (TCA) cycle, such as citrate, isocitrate, alpha-ketoglutarate, and malic acid declined along the berry development in RR (Figure 2B). Levels of most of the detected amino acids increased while those of several key stress responsive amino acids (such as 2-aminoadipic acid, aspartate, glutamine, and serine) and glutamate metabolism amino acids (such as ornithine, arginine, and pyroglutamic acid) declined, to different extents, in RR (Figure 2B). Levels of 50% lipids declined at early stages while several other lipids (including lysoPEs, acetoacetate, and phosphocholine) increased from veraison in RR. Levels of 50% nucleotides accumulated more in RR (Appendix A).

For secondary metabolites, as compared with those in RC, levels of 18 flavonoids (including five anthocyanins, six flavanols, two flavanones, and five flavones), two cinnamic acids, two benzene and substituted derivatives, one hydrolysable tannins, and three others kept constitutively higher at all berry development, while those of 21 flavonoids (including three anthocyanins, six flavones, and 12 flavanols) (Appendix A), three cinnamic acids, two benzene and substituted derivatives, four hydrolysable tannins, and one other kept constitutively lower in RR (Figure 3A, Appendix A). Among the higher level metabolites, the ratios peaked at S2, and then sharply declined to two to four folds at S5. Levels of seven flavonoids, six cinnamic acids, and three benzene and substituted derivatives were significantly lower in RR at early stage, and then kept constant (Appendix A). In addition, levels of 11 secondary metabolites (isovitexin, morin, delphinidin-3-*O*-(6′′-*O*-alpha-rhamnopyranosyl-beta-glucopyranoside), galloyl-HHDP-gluc-ose#1, syringic acid, protocatechuic acid, 3-cresotinic acid, 2-Amino-1,3,4-tetradecanetriol, tetrabutylammonium, *N*,*N*′-Dicyclohexylurea, dodecyl (dimethyl) amine oxide) did not change differentially at all berry development (Appendix A). 

## 3. Discussion

Grape berries at different phenological stages have their special metabolomes, which is prone to be influenced by both biotic and abiotic stresses. It is reported that variations in metabolic compositions affected by environmental perturbations will be associated with final yield at harvest [38]. Therefore, understanding of the metabolic changes in the metabolomics level in developing grape berries is of remarkable significance regarding the yield and quality improvement of grape berries, particularly under stress conditions. In this study, we investigated into the kinetic changes of the metabolites of grape berry skin tissues in response to RR along grape berry development using non-targeted metabolomics. Our results showed that the metabolome of grape berry skin is development-stage dependent, and that general kinetics of metabolic changes in developing grape skin tissues are similar between RR and RC samples, which indicated a highly conserved metabolic regulatory mechanism of grape berry development. Our metabolomics data, however, revealed systematic influences of RR on skin metabolome, that could be used to explain the observed differences in the flavor and taste of grape berry in RR samples. 

### 3.1. Effect of RR on Primary Metabolism in Skin Tissues of Developong Berries 

#### 3.1.1. Amino Acids

Previous studies have reported that levels of compounds related to nitrogen uptake, assimilation, and transport decrease in various vegetative organs (including root, cane, truck, shoot, leaf as well as xylem sap) under RR condition [24]. Our study found that nearly 63% annotated amino acids increased in reproductive organ berry skin tissues at early stage of berry development (S1). This could be the differential responses of vegetative and reproductive tissues to RR, alternatively, it could be the result of natural evolution, in which grapevine tends to prioritize reproductive growth under stress condition. For example, the level of proline was significantly higher than that in RC. Proline, an osmotic adjustment substance, is the main amino acid for storing carbon and nitrogen in grapevine, and its accumulation is considered to be a physiological response of plants to biotic and abiotic stresses [39]. Therefore, the increase of proline in RR berry skin tissues could be an alternative biomarker of the adaptive ability of grapevine plant in response to RR stress [40]. Similarly, glutamate, one of the precursor of proline, increased dramatically in RR at early stage (S1–S2), which benefited the increase of proline [41]. In addition, levels of aromatic amino acids (such as phenylalanine, tryptophan, and tyrosine) increased dramatically in RR, which indicated a remarkable metabolic shift from primary metabolism to secondary metabolism in skin tissues of developing berries in RR sample, because all those aromatic amino acids are important precursors for secondary metabolism. The significant increases of aromatic amino acids could contribute to the improved berry color and taste in berries under RR condition. On the other hand, abundances of several key stress responsive amino acids, such as 2-aminoadipic acid, aspartate, glutamine, and serine declined gradually to a larger extent in RR at almost all tested stages (Figure 2). It is reported that those stress responsive amino acids are consumed for sugar and energy metabolism, or precursors of other essential amino acids [42]. Accompanied with the declining of glutamate in RR along berry development, levels of several other amino acids (such as ornithine, arginine, and pyroglutamic acid) that were related to the glutamate metabolism [43], decreased as well to a large extent in RR as compared with those in RC. Altogether, RR exerted significant effects on amino acid metabolism in skin tissues of developing berries for adaptive response to RR stress. 

#### 3.1.2. Lipids

Lipids are one of important class of primary metabolites, which are known to be involved in many biological processes, such as being structural components of cell membranes as phospholipid bilayers, participating in signal transport, and providing structural and functional molecules for energy metabolism [44]. Changes in lipid metabolism can also be seen as the indicators of environmental stresses [45,46]. Root restriction as an environment stress resulted in the accumulation of higher contents of lipids, especially lysoPEs, in skin tissues of late stage berries (Appendix A). It is well known that lysoPEs are sensitive to environment changes, because most of them function as cell-mediate signaling molecules or special enzymes [47]. Therefore, our results indicated that RR cultivation may employ a special lipid metabolism strategy as a potential biological mechanism in berry skin against the environment variation. 

#### 3.1.3. Carbohydrates, Organic Acids, and Other Primary Metabolites.

Generally, the ratio of sugar to organic acids determines the quality of fruit. Previous reports have proved that higher total soluble sugar content in berry was induced by high acid invertase (AI) activity under RR [25]. In this study, however, we did not observe obvious increases of soluble sugars in skin tissues of berries under RR condition as compared with those in RC, instead, we observed stable levels of fructose and sucrose along the berry development, except that glucose did increase significantly in skin tissues of developing berries under RR condition. This discrepancy is likely due to the fact that berry skin is the main site for secondary metabolites accumulation [10,11], which has different metabolic profile from that of the berry pulp. On the other hand, the significant decline of TCA intermediates (including organic acids) was found in skin tissues of berries at pre-veraison or veraison stage under RR condition (Figure 2). Similar results were found in grape of Cabernet Sauvignon and Merlot [1]. The declined organic acids in RR samples would facilitate amino acids biosynthesis as mentioned above to respond to RR stress. Among other primary metabolites, levels of nucleotides accumulated more in RR than that in RC, which could be a similar consequence of a general response of plant to other stresses.

### 3.2. Effect of RR on Secondary Metabolism in Skin Tissues of Developing Berries 

In response to water stress, the expression pattern of grape berry at transcriptional level was tissue-specific, and the skin appeared to undergo more pronounced changes in transcriptome profiling than that of pulp tissue [48]. It is reported that RR may cause water stress to the plant, resulting in the accumulation of antioxidants and flavor compounds, especially phenolics [24,37]. In this study, RR significantly increased levels of three monoterpenol glycoconjugations and most of flavonoids in skin tissues of developing berries from pre-veraison (S2) (Figure 3). The increased flavonoids in berry skin tissues under RR condition could be the direct response of berry skin to RR stress, providing antioxidants to protect berries as suggested in many other plant systems [49]. This result likely comes from the upregulation of genes involved in flavonoids metabolism as reported previously [50]. However, levels of metabolites in the flavanols pathway, a sub-branch of flavonoids, were found to be considerably declined along berry development; these metabolites included epicatechin, epigallocatechin, and procyanidin dimer B1–B5 (Figure 3 and Appendix A). This observation was consistent with the previous studies that flavanols are principally enriched in young and developing tissues [51]. It is worthy to note that abundances of flavonoids determine the color and astringency of grape, which is an key index of berry quality, and it is reported that RR induce anthocyanin accumulation in berry skin [22,27]. In this study, our metabolomics data not only confirmed the accumulation of anthocyanins in RR samples, but also showed that RR induced accumulation of other flavonoids such as flavones and flavanols in berry skin tissues. Therefore, RR could be applied for practical improvement of flavor and taste of berry.

Notably, most flavonoids accumulated more rapidly at pre-veraison (S2) stage in skin tissues in RR than that in RC. This could be the reason why grape berry in RR advanced earlier into veraison stage than that in RC [30]. The advanced coloring in RR skin tissues is consistent with the previous finding that the large amount of flavanols synthesized at early stage contributed to the increased total flavonoids content in skin tissue and to the acceleration of grape ripening [52]. In addition, flavonoids are reported to function similarly to that of the synthetic auxin transport inhibitor naphthylphthalamic acid (NPA), acting as inhibitors of auxin transport [53]. Our study found that RR increased flavonoids accumulation accompanied with advanced coloring, indicated that RR inhibits grapevine vegetative growth while promotes reproductive growth, which was consist with previous studies [23,40]. Thus, RR is a potential alternative method for production practice. 

## 4. Materials and Methods 

### 4.1. Plant Material and Sample Collection

Three-year-old “Red Alexandria” table grape vines were planted with RR treatments or with RC in the greenhouse of Shanghai Jiao Tong University (31°11′ N, 121°29′ E), Shanghai, China, during the 2016–2017 growing year. The grape vines for RR group were grown in plastic boxes (60 cm × 45 cm × 45 cm), while those for RC were grown in a 45-cm deep raised bed at open ground. Both RR and RC vines were cultivated with the same medium (a mixture of sand, loam, and manure, 1:1:1, *v*/*v*/*v*), under the same watering and fertilizer conditions. 

Grape berries at five developmental stages, namely S1 (hard-core stage, eight weeks after full bloom, WAFB), S2 (pre-veraison stage, ten WAFB), S3 (veraison stage, 12 WAFB), S4 (pre-ripening stage, 14 WAFB), and S5 (harvest-ripe stage, 16 WAFB) were collected. At each sampling phase, 10 clusters without any evidence of stress or disease symptoms from at least 5 individual vines were randomly collected. Three biological replicates were performed for both RR and RC. Each biological replicate contained 10 berries randomly selected from a pool of berries harvested at each sampling stage. All berries’ skin tissues were separated from pulp tissues as soon as possible upon harvest, then frozen in liquid nitrogen immediately, ground into power in liquid nitrogen, and stored at −80 °C.

### 4.2. Metabolite Extraction and Profiling

Skins tissue materials of 50 mg were extracted with 40 volumes (*w*/*v*) of ice cold methanol [54]. The extracts were vortexed and sonicated in an ultrasonic bath at room temperature for 30 min at 40 Hz. The crude extracts were centrifuged for 10 min at 12,000 rpm at 4 °C, and the supernatants were filtered with pore filter (0.2 μm diameter pores) before injecting into the UHPLC-MS system.

The UHPLC analysis was carried out using an Agilent 1290 Infinity II LCTM system, equipped with an Agilent Eclipse-plus C18 column (150 × 3.0 mm, particle size 1.8 μm), and the column temperature was set to 40 °C. Two mobile phase agents were used: 0.1% formic acid in water (solvent A) and 100% acetonitrile (solvent B). The LC gradient conditions were as follows: 98% A during 0–1 min, a linear decrease from 98% to 60% A during 1–5 min, from 60% to 30% A during 5–12 min, and from 30% to 5% A during 12–15 min, finally 5% A during 15–20 min with a 100-μL injection volume and a flow rate of 0.4 mL/min. Both positive and negative ionization spectra were recorded in a range of 50–1000 *m*/*z* (full scan spectra mode with a scan rate of 2 spectra/s). Sixteen L/min for drying gas and 25 psi for nebulizer heated at 350 °C. Additional parameters: a capillary voltage of 3500 V (+), nozzle voltage at 1.5 kV (−), and 250 V (+), fragmentor voltages at 380, 10, 20, and 40 V were applied to collision-induced dissociation (CID) voltage. 

Metabolite detection was performed with an Agilent 6550 iFunnel/Q-TOF mass spectrometer, equipped with an Agilent Jet-Stream source. Detailed information regarding data acquisition can be found in a previous study [55]. Metabolite annotations were done by searching Personal Compound Database and Library (PCD/PCDL), Massbank database (http://www.massbank.jp/en/manual.html) [56], Metlin database (http://metlin.scripps.edu) [57] and data reported in literature [34]. MassHunter Acquisition 6.0, MassHunter Qualitative 6.0, and Mass Profinder 6.0 were used for data acquisition, review, and peak area extraction, respectively. For data processing, Masshunter qual software (Agilent, Santa Clara, CA, USA) was used for identifying the compounds and Batch Targeted Feature Extraction function in Mass Hunter Profinder 6.0 (Agilent) was used for peak area extraction. [M + H]^+^ and [M + Na]^+^ adducts were recorded for the metabolites identified in the positive mode, while [M − H]^−^ adducts were recorded for most of the metabolites identified in the negative mode except for lysoPCs (which were [M + HCOO]^−^ adducts). The extracted ion chromatography peak integration and filtering: integrator selection was Agile 2, selected smooth EIC before integration and selected Gaussian as smoothing function, nine points and 5.000 points was set as function width and Gaussian width, respectively. Then peak height filters with absolute height ≥500 counts. After the automatic extraction of peak area, each metabolite was carefully reviewed. If the automatic chromatographic integration was wrong, then manual integration was performed.

### 4.3. Statistical Analysis

The data normalization was carried out as reported previously [55]. The peak areas were used to quantify the abundance of metabolites, which were divided by the median value of each metabolite and the weight of sample. The data were analyzed by principle component analysis (PCA) using SIMCA-P version 11.0. False discovery rate (FDR) was used for multiple testing correlation (FDR < 0.05) [55]. Welch’s two-sample *t*-test (*p* < 0.05) was applied to all comparisons between different groups. The heatmaps of metabolites ratios were generated using MultiExperiment Viewer version 4.8.

## 5. Conclusions

In this study, we successfully annotated 291 metabolites in developing berry skin tissues using a non-targeted metabolomics approach, and compared the metabolomics kinetics of developing berry skin tissues under RR and RC conditions. To our knowledge, this is the first comprehensive metabolomics report investigating the dynamic metabolic changes in berry skin tissues of developing berries in response to RR. An earlier and more rapid metabolic shift from primary metabolism to secondary metabolism was found in RR samples which were accompanied with an advanced coloring of the berries, as compared with that in RC ones, indicating that RR could be a useful production practice for improvement of berry appearance and taste quality. This study expands our understanding of berry development and quality improvement. Further studies will focus on the validation of those metabolic findings using either targeted or combined non-targeted methods, and on the molecular and genetic elements underlying those differential metabolomics changes occurred in RR and RC skins. 

## Figures and Tables

**Figure 1 ijms-20-00534-f001:**
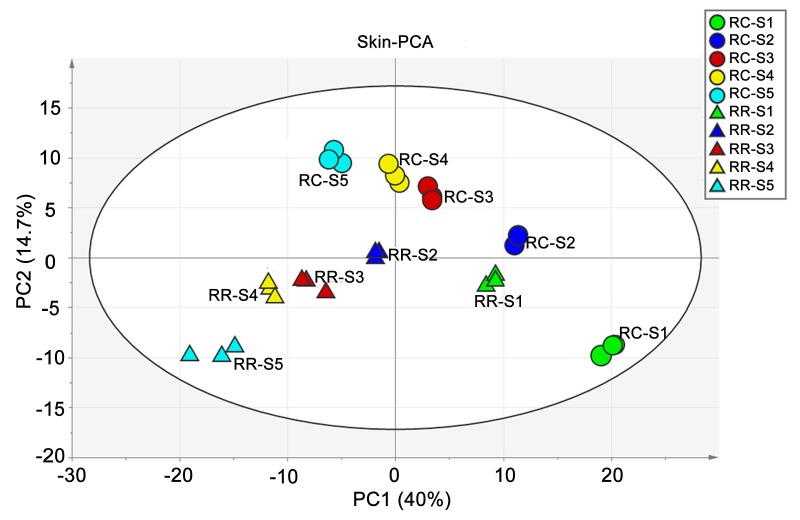
Principal component analysis (PCA) of the metabolites annotated in developing grape berry skin. Circle and triangle donate traditional cultivation (RC) and root restriction (RR) cultivation method, respectively. Green, blue, brown, yellow, and cyan colors represent samples collected at S1, S2, S3, S4, S5 stage, respectively. S1, eight weeks after full bloom (WAFB); S2, ten WAFB; S3, 12 WAFB; S4, 14 WAFB; S5, 16 WAFB (berry maturation). Principle Component (PC) 1 explains 40% of variance distinguishing grape berry skin samples from different developmental stages. PC2 explains 14.7% of variance distinguishing skin samples from different cultivation methods.

**Figure 2 ijms-20-00534-f002:**
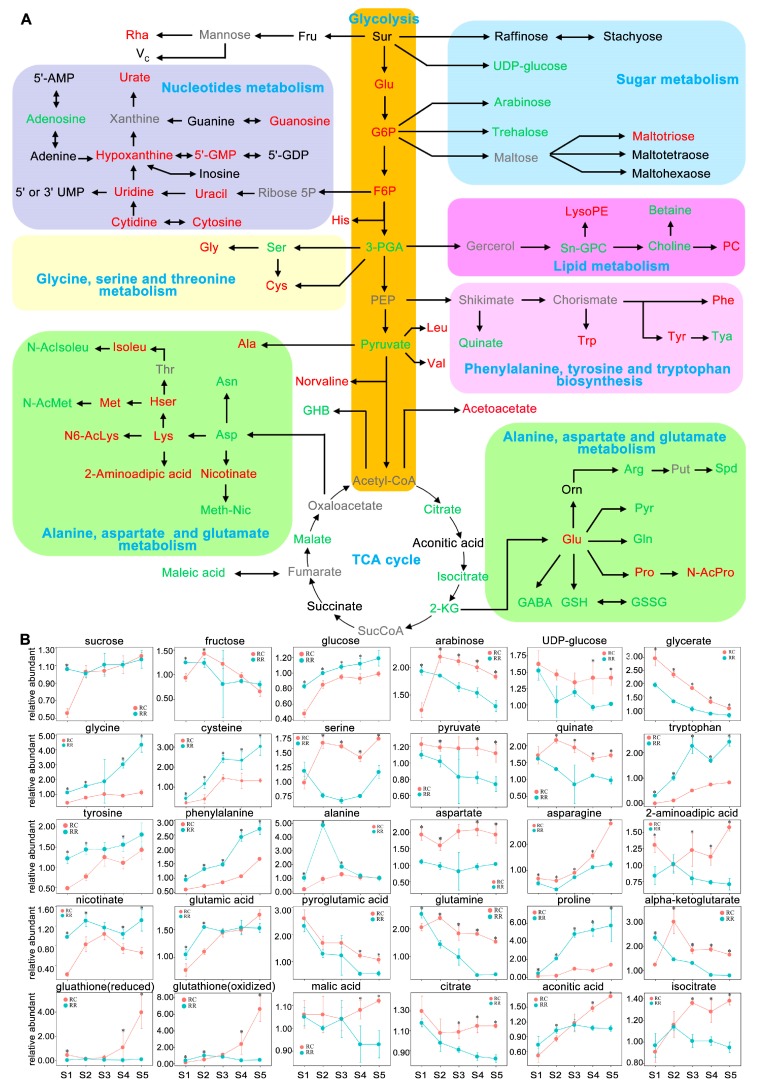
(**A**) Visualization of metabolic dynamics of primary metabolism in skin tissues of developing berries. Representative metabolites with significantly statistical increases and declines more than three stages in RR are written in red and green letters, respectively. Metabolites with black letters indicate their dynamic changes were not significantly different between RR and RC. Metabolites with grey letters represent undetected metabolites. (**B**) Partial of representative metabolites. The relative levels of metabolites were averaged with three biological replicates. Bars represent standard errors. Significant differences were signaled by an asterisk (*p* < 0.05) using *t*-test. Each color of background represents one metabolic pathway.

**Figure 3 ijms-20-00534-f003:**
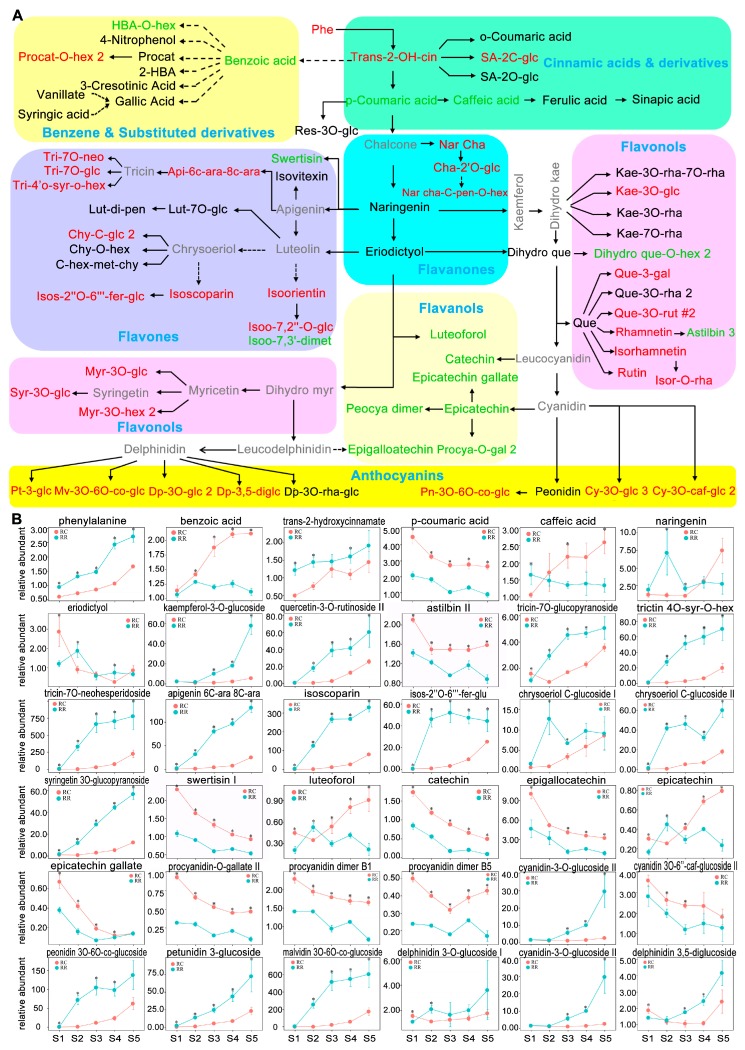
(**A**) Visualization of metabolic dynamics of secondary metabolism in skin tissues of developing grape berries. Representative metabolites with significantly statistical increases and declines more than three stages in RR are written in red and green letters, respectively. Metabolites with black letters indicate their dynamic changes were not significantly different between root restriction (RR) and traditional cultivation (RC). Metabolites with grey letters represent undetected metabolites. (**B**) Partial of representative metabolites are presented. The relative levels of metabolites were averaged with three biological replicates. Bars represent standard errors. Significant differences were signaled by an asterisk (*p* < 0.05) using *t*-test. Each color of background represents one subclass of phenolic pathway.

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
