# Peer review of "Comparative Metabolic Profiling of Grape Skin Tissue along Grapevine Berry Developmental Stages Reveals Systematic Influences of Root Restriction on Skin Metabolome"

_ijms, 2019, doi:10.3390/ijms20030534_

Round 1

Reviewer 1 Report

Dear Authors and Editors,

I reviewed manuscript "Comparative metabolic profiling of grape skin tissue along grapevine berry developmental stages reveals systematic influences of root restriction on skin metabolome" submitted for publication to International Journal of Molecular Sciences.

In general, untargeted metabolomic analysis is very useful method but crucial is data processing after data acquisition, especially when LC-MS/MS systems have been used. In a reviewed manuscript authors mentioned that they used Mass Hunter Profinder 6.0 but without details which adducts they recorded. Adducts are always present when is plant material analyzed. Please explain data processing in details. Which parameters in Mass Hunter Profinder did you use?

Untargeted analysis gave big set of data and it is very important to present data on clear and understandable way. In my opinion authors did not achieve this in reviewed manuscript. They have two variables which they want to study- development stage and cultivation method. The whole manuscript would be clearer if authors would present just one variable. Now, in some parts it is hard to understand what authors compare. They actually show the same results in a few different ways. In my opinion, just Figure 6. and  7. are enough to explain what authors did. But these Figures should be bigger and resolution should be better. Now these Figures are blurry and hard to read. Please rewrite results section.

L 94-99 there is no need to present results of the work in introduction section

Please remove Figure 2; they do not bring any new evidence what is not evident from PCA. All what authors explained here can be explained from PCA.

Heatmaps at Figure 3. and 4. very confusing. Please, create one heatmap or present data somehow different. Figures should be clear by themselves and these Figures are very hard to understand.

2.5. Metabolite-metabolite correlation analysis totally unnecessary and should be removed.

Author Response

Comment 1: In general, untargeted metabolomic analysis is very useful method but crucial is data processing after data acquisition, especially when LC-MS/MS systems have been used. In a reviewed manuscript authors mentioned that they used Mass Hunter Profinder 6.0 but without details which adducts they recorded. Adducts are always present when is plant material analyzed. Please explain data processing in details. Which parameters in Mass Hunter Profinder did you use?

Answer: Thanks for this critical comment. We have added detailed description of the data processing using Mass Hunter Profinder 6.0 in the Materials and Methods section (Page 10 line 335 to page 11 line 345).

Comment 2: Untargeted analysis gave big set of data and it is very important to present data on clear and understandable way. In my opinion authors did not achieve this in reviewed manuscript. They have two variables which they want to study- development stage and cultivation method. The whole manuscript would be clearer if authors would present just one variable. Now, in some parts it is hard to understand what authors compare. They actually show the same results in a few different ways. In my opinion, just Figure 6. and 7. are enough to explain what authors did. But these Figures should be bigger and resolution should be better. Now these Figures are blurry and hard to read. Please rewrite results section.

Answer: Thank you very much for this constructive comment. We have revised the Results section thoroughly. We first revealed the metabolic variation of developing berries to reflect the phenological variations of developing berries, which is stage-dependent. We second compared the effect of RR and RC on berry metabolome, and revealed the metabolic effect of RR on berry skin color and quality.

Comment 3: L 94-99 there is no need to present results of the work in introduction section

Answer: We have deleted this as suggested.

Comment 4: Please remove Figure 2; they do not bring any new evidence what is not evident from PCA. All what authors explained here can be explained from PCA.

Answer: The original Figure 2 has been deleted as suggested.

Comment 5: Heatmaps at Figure 3. and 4. very confusing. Please, create one heatmap or present data somehow different. Figures should be clear by themselves and these Figures are very hard to understand.

Answer: We have moved the original two heatmaps to the supplementary Section.

Comment 6: 2.5. Metabolite-metabolite correlation analysis totally unnecessary and should be removed.

Answer: We have deleted this as suggested. 

Reviewer 2 Report

The paper presents interesting results concerning the effect of root restriction on the metabolome of grapes skin. Future employment of other metabolomics platform could confirm these results and this consideration might also be included in the conclusions.

Although interesting, the paper needs major language revision. It seems at times that it was not even re-read and words are in some cases missing. Grammar and structure are in fact pretty poor at this moment and many typos are present in the text. I highly recommend the help of an English native to restructure most sentences and check the spelling and grammar. 

One minor comment would also be that if RR is introduced as an abbreviation, then it must be used consistently throughout the article.

After the provision of a new, corrected version of the paper, it could be considered for acceptance, in my opinion.

Author Response

Comment 1: The paper presents interesting results concerning the effect of root restriction on the metabolome of grapes skin. Future employment of other metabolomics platform could confirm these results and this consideration might also be included in the conclusions.

Answer: We have revised the conclusion as suggested (Page 11 line 364-367).

Comment 2: Although interesting, the paper needs major language revision. It seems at times that it was not even re-read and words are in some cases missing. Grammar and structure are in fact pretty poor at this moment and many typos are present in the text. I highly recommend the help of an English native to restructure most sentences and check the spelling and grammar. 

Answer: We appreciated very much for this helpful comment. We have revised the MS thoroughly and asked one colleague who is native English speaker to help us to improve the English writing.

Comment 3: One minor comment would also be that if RR is introduced as an abbreviation, then it must be used consistently throughout the article.

Answer: We have changed as suggested. 

Round 2

Reviewer 1 Report

Dear Authors,

Now manuscript is much better, easy to follow and presentation is clearer.

Author Response

Thank you very much for your constructive comments to previous manuscript.